# PRINCIPLED ARCHITECTURE-AWARE SCALING OF HYPERPARAMETERS

**Wuyang Chen**[1] , **Junru Wu**[2*], **Zhangyang Wang**[1], **Boris Hanin**[3]
[1]University of Texas, Austin    [2]Google Research    [3]Princeton University
[1]{wuyang.chen,atlaswang}@utexas.edu    [2]junru@google.com
[3]bhanin@princeton.edu

## ABSTRACT

Training a high-quality deep neural network requires choosing suitable hyperparameters, which is a non-trivial and expensive process. Current works try to automatically optimize or design principles of hyperparameters, such that they can generalize to diverse unseen scenarios. However, most designs or optimization methods are agnostic to the choice of network structures, and thus largely ignore the impact of neural architectures on hyperparameters. In this work, we precisely characterize the dependence of initializations and maximal learning rates on the network architecture, which includes the network depth, width, convolutional kernel size, and connectivity patterns. By pursuing every parameter to be maximally updated with the same mean squared change in pre-activations, we can generalize our initialization and learning rates across MLPs (multi-layer perception) and CNNs (convolutional neural network) with sophisticated graph topologies. We verify our principles with comprehensive experiments. More importantly, our strategy further sheds light on advancing current benchmarks for architecture design. A fair comparison of AutoML algorithms requires accurate network rankings. However, we demonstrate that network rankings can be easily changed by better training networks in benchmarks with our architecture-aware learning rates and initialization. Our code is available at `https://github.com/VITA-Group/principled_scaling_lr_init`.

## 1 INTRODUCTION

An important theme in the success of modern deep learning is the development of novel neural network architectures: ConvNets (Simonyan & Zisserman, 2014), ResNets (He et al., 2016), Transformers (Dosovitskiy et al., 2020) to name a few. Whether a given architecture works well in practice, however, depends crucially on having reliable principles for selecting training hyperparameters. Indeed, even using more or less standard architectures at scale often requires considerable hyperparameter tuning. Key examples of such hyperparameters are the **learning rate** and the network **initialization**, which play a crucial role in determining the quality of trained models. How to choose them in practice depends non-trivially on the interplay between data, architecture, and optimizer.

**Principles** for matching deep networks to initialization and optimization schemes have been developed in several vibrant lines of work. This includes approaches — typically based on analysis in the infinite width limit — to developing initialization schemes suited to specific architectures, including fully connected architectures (Jacot et al., 2018; Perrone et al., 2018), ConvNets (Ilievski & Feng, 2016; Stoll et al., 2020), ResNets (Horváth et al., 2021; Li et al., 2022b;a), and Transformers (Zhu et al., 2021; Dinan et al., 2023). It also includes several approaches to zero-shot hyperparameter transfer, which seek to establish the functional relationships between a hyperparameter (such as a good learning rate) and the architecture characteristics, such as depth and width. This relation then allows for hyperparameter tuning on small, inexpensive models, to be transferred reliably to larger architectures (Yang et al., 2022; Iyer et al., 2022; Yaida, 2022).

**Architectures** of deep networks, with complicated connectivity patterns and heterogeneous operations, are common in practice and especially important in neural architectures search (Liu et al., 2018;

---

*Work done while at Texas A&M University

Dong & Yang, 2020). This is important because long-range connections and heterogeneous layer types are common in advanced networks (He et al., 2016; Huang et al., 2017; Xie et al., 2019). However, very little prior work develops initialization and learning rate schemes adapted to complicated network architectures. Further, while automated methods for hyperparameter optimization (Bergstra et al., 2013) are largely model agnostic, they cannot discover generalizable principles. As a partial remedy, recent works (Zela et al., 2018; Klein & Hutter, 2019; Dong et al., 2020b) jointly design networks and hyperparameters, but still ignore the dependence of learning rates on the initialization scheme. Moreover, ad-hoc hyperparameters that are not tuned in an intelligent way will further jeopardize the design of novel architectures. Without a fair comparison and evaluation of different networks, benchmarks of network architectures may be misleading as they could be biased toward architectures on which "standard" initializations and learning rates we use happen to work out well.

This article seeks to develop **principles** for both initialization schemes and zero-shot learning rate selection for general neural network **architectures**, specified by a directed acyclic graph (DAG). These architectures can be highly irregular (Figure 1), significantly complicating the development of simple intuitions for training hyperparameters based on experience with regular models (e.g. feedforward networks). At the beginning of gradient descent training, we first derive a simple initialization scheme that preserves the variance of pre-activations during the forward propagation of irregular architectures, and then derive architecture-aware learning rates by following the maximal update ($\mu$P) prescription from (Yang et al., 2022). We generalize our initialization and learning rate principles across MLPs (multi-layer perception) and CNNs (convolutional neural network) with arbitrary graph-based connectivity patterns. We experimentally verify our principles on a wide range of architecture configurations. More importantly, our strategy further sheds light on advancing current benchmarks for architecture design. We test our principles on benchmarks for neural architecture search, and we can immediately see the difference that our architecture-aware initializations and architecture-aware learning rates could make. Accurate network rankings are required such that different AutoML algorithms can be fairly compared by ordering their searched architectures. However, we demonstrate that network rankings can be easily changed by better training networks in benchmarks with our architecture-aware learning rates and initialization. Our main contributions are:

- We derive a simple modified fan-in initialization scheme that is architecture-aware and can provably preserve the flow of information through any architecture's graph topology (§ 3.2).
- Using this modified fan-in initialization scheme, we analytically compute the dependence on the graph topology for how to scale learning rates in order to achieve the *maximal update* ($\mu$P) heuristic (Yang et al., 2022), which asks that in the first step of optimization neuron pre-activations change as much as possible without diverging at large width. For example, we find that (§ 3.3) for a ReLU network of a graph topology trained with MSE loss under gradient descent, the $\mu$P learning rate scales as $\left(\sum_{p=1}^{P} L_p^3\right)^{-1/2}$, where $P$ is the total number of end-to-end paths from the input to the output, $L_p$ is the depth of each path, $p = 1, \cdots, P$.
- In experiments, we not only verify the superior performance of our prescriptions, but also further re-evaluate the quality of standard architecture benchmarks. By unleashing the potential of architectures (higher accuracy than the benchmark), we achieve largely different rankings of networks, which may lead to different evaluations of AutoML algorithms for neural architecture search (NAS).

## 2 RELATED WORKS

### 2.1 HYPERPARAMETER OPTIMIZATION

The problem of hyperparameter optimization (HPO) has become increasingly relevant as machine learning models have become more ubiquitous (Li et al., 2020; Chen et al., 2022; Yang et al., 2022). Poorly chosen hyperparameters can result in suboptimal performance and training instability. Beyond the basic grid or random search, many works tried to speed up HPO. (Snoek et al., 2012) optimize HPO via Bayesian optimization by jointly formulating the performance of each hyperparameter configuration as a Gaussian process (GP), and further improved the time cost of HPO by replacing the GP with a neural network. Meanwhile, other efforts tried to leverage large-scale parallel infrastructure to speed up hyperparameter tuning (Jamieson & Talwalkar, 2016; Li et al., 2020).

Our approach is different from the above since we do not directly work on hyperparameter optimization. Instead, we analyze the influence of network depth and architecture on the choice of hyperparameters. Meanwhile, our method is complementary to any HPO method. Given multiple models, typical methods optimize hyperparameters for each model separately. In contrast, with our method, one only needs to apply HPO **once** (to find the optimal learning rate for a small network), and then directly scale up to larger models with our principle (§ 4).

## 2.2 Hyperparameter Transfer

Recent works also tried to propose principles of hyperparameter scaling and targeted on better stability for training deep neural networks (Glorot & Bengio, 2010; Schoenholz et al., 2016; Yang & Schoenholz, 2017; Zhang et al., 2019; Bachlechner et al., 2021; Huang et al., 2020; Liu et al., 2020). Some even explored transfer learning of hyperparameters (Yogatama & Mann, 2014; Perrone et al., 2018; Stoll et al., 2020; Horváth et al., 2021). (Smith et al., 2017; Hoffer et al., 2017) proposed to scale the learning rate with batch size while fixing the total epochs of training. (Shallue et al., 2018) demonstrated the failure of finding a universal scaling law of learning rate and batch size across a range of datasets and models. Assuming that the optimal learning rate should scale with batch size, (Park et al., 2019) empirically studied how the optimal ratio of learning rate over batch size scales for MLP and CNNs trained with SGD.

More recently, Yang & Hu (2020) found that standard parameterization (SP) and NTK parameterization (NTP) lead to bad infinite-width limits and hence are suboptimal for wide neural networks. Yang et al. (2022) proposed $\mu$P initialization, which enabled the tuning of hyperparameters indirectly on a smaller model, and zero-shot transfer them to the full-sized model without any further tuning. In (Iyer et al., 2022), an empirical observation was found that, for fully-connected deep ReLU networks, the maximal initial learning rate follows a power law of the product of width and depth. Jelassi et al. (2023) recently studied the depth dependence of $\mu$P learning rates vanilla ReLU-based MLP networks. Two core differences between $\mu$P and our method: 1) $\mu$P mainly gives width-dependent learning rates in the first and last layer, and is width-independent in other layers (see Table 3 in (Yang et al., 2022)). However, there is a non-trivial dependence on both network depth and topologies; 2) Architectures studied in $\mu$P mainly employ sequential structures without complicated graph topologies.

## 2.3 Bechmarking Neural Architecture Search

Neural architecture search (NAS), one research area under the broad topic of automated machine learning (AutoML), targets the automated design of network architecture without incurring too much human inductive bias. It is proven to be principled in optimizing architectures with superior accuracy and balanced computation budgets (Zoph & Le, 2016; Tan et al., 2019; Tan & Le, 2019). To fairly and efficiently compare different NAS methods, many benchmarks have been developed (Ying et al., 2019; Dong & Yang, 2020; Dong et al., 2020a). Meanwhile, many works explained the difficulty of evaluating NAS. For example, (Yang et al., 2019) emphasized the importance of tricks in the evaluation protocol and the inherently narrow accuracy range from the search space. However, rare works point out one obvious but largely ignored defect in each of these benchmarks and evaluations: diverse networks are blindly and equally trained under the same training protocol. It is questionable if the diverse architectures in these benchmarks would share the same set of hyperparameters. As a result, to create a credible benchmark, it's necessary to find optimal hyperparameters for each network accurately and efficiently, which our method aims to solve. This also implied, NAS benchmarks that were previously regarded as "gold standard", could be no longer reliable and pose questions on existing NAS algorithms that evaluated upon it.

## 3 Methods

For general network architectures of directed acyclic computational graphs (DAGs), we propose topology-aware initialization and learning rate scaling. Our core advantage lies in that we **precisely characterize** the non-trivial dependence of learning rate scales on both networks' depth, DAG topologies and CNN kernel sizes, beyond previous heuristics, with practical implications (§ 3.5).

1. In a DAG architecture (§ 3.1), an initialization scheme that depends on a layer's in-degree is required to normalize inputs that flow into this layer (§ 3.2).

2. Analyzing the initial step of SGD allows us to detail how changes in pre-activations depend on the network's depth and learning rate (§ B.1.

3. With our in-degree initialization, by ensuring the magnitude of pre-activation changes remains $\Theta(1)$, we deduce how the learning rate depends on the network topology (§ 3.3) and the CNN kernel size (§ 3.4).

## 3.1 DEFINITION OF DAG NETWORKS

We first define the graph topology of complicated neural architectures. This formulation is inspired by network structures designed for practical applications (Xie et al., 2019; You et al., 2020; Dong & Yang, 2020).

By definition, a directed acyclic graph is a directed graph $\mathcal{G} = (V, E)$ in which the edge set has no directed cycles (a sequence of edges that starts and ends in the same vertex). We will denote by $L + 2 := |V|$ the number of vertices (the input $x$ and pre-activations $z$) in $V$ and write:

$$V = [0, L+1] := \{0, 1, \ldots, L+1\}.$$

We will adopt the convention that edges terminating in vertex $\ell$ originate only in vertices $\ell' < \ell$. We define a DAG neural network (with computational graph $\mathcal{G} = (V, E)$, the number of vertices $|V| = L + 2$, hidden layer width $n$, output dimension 1, and

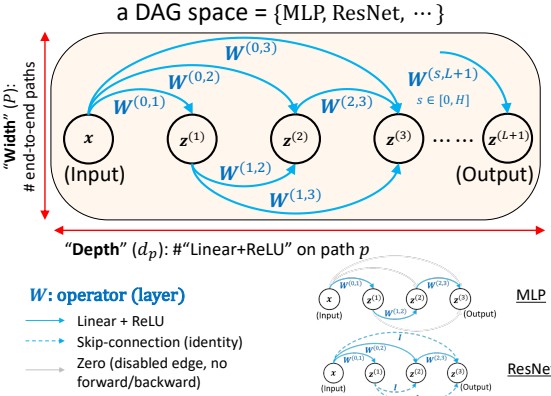

Figure 1: A neural network's architecture can be represented by a direct acyclic graph (DAG). $x$ is the input, $z^{(1)}, z^{(2)}, \cdots, z^{(L)}$ are pre-activations (vertices), and $z^{(L+1)}$ is the output. $W$ is the layer operation (edge). Our DAG space includes architectures of different connections and layer types ("Linear + ReLU", "Skip-connect", and "Zero"). For example, by disabling some edges (gray) or replacing with identity $I$ (skip connection, dashed arrow), a DAG can represent practical networks such as MLP and ResNet (He et al., 2016).

ReLU activations) to be a function $x \in \mathbb{R}^n \mapsto z^{(L+1)}(x) \in \mathbb{R}$ given by

$$z^{(\ell)}(x) = \begin{cases} x, & \ell = 0 \\ \sum_{(\ell', \ell) \in E} W^{(\ell', \ell)} \sigma(z^{(\ell')}(x)), & \ell = 1, \ldots, L+1 \end{cases}, \tag{1}$$

$\sigma(t) = \mathrm{ReLU}(t) = \max\{0, t\}$. $W^{(\ell', \ell)}$ are $n \times n$ matrices of weights initialized as follows:

$$W_{ij}^{(\ell', \ell)} \sim \begin{cases} \mathcal{N}(0, C^{(\ell', \ell)}/n), & \ell = 1, \ldots, L \\ \mathcal{N}(0, C^{(\ell', \ell)}/n^2), & \ell = L+1 \end{cases}, \tag{2}$$

where $C^{(\ell', \ell)}$ is a scaling variable that we will determine based on the network topology (equation 4). We choose $1/n^2$ as the variance for the output layer mainly for aligning with $\mu$P's desiderata.

## 3.2 TOPOLOGY-AWARE INITIALIZATION SCHEME

A common way to combine features from multiple layers is to directly sum them up, often without normalization (He et al., 2016; Dong & Yang, 2020) as we do in equation 1. This summation implies that the weight magnitude of each pre-activation layer should scale proportionally regarding the number of inputs (in-degree). However, most commonly used weight initialization schemes, such as He initialization (He et al., 2015) and Xavier initialization (Glorot & Bengio, 2010), only consider forward or backward propagation between consecutive layers, ignoring any high-level structures, i.e. how layers are connected with each other (see also (Defazio & Bottou, 2022)). This strategy can be problematic in the presence of a bottleneck layer that has a large number of inputs.

Motivated by this issue, we propose an architecture-aware initialization strategy. We target deriving the normalizing factor $C^{(\ell', \ell)}$ in the initialization (equation 2) adapted to potentially very different in-degrees of the target layer. Like many theoretical approaches to neural network initialization, we do this by seeking to preserve the "flow of information" through the network at initialization. While this dictum can take various guises, we interpret it by asking that the expected norm of changes in pre-activations between consecutive layer have the same typical scale:

$$\mathbb{E}\left[\left\|z^{(\ell)}\right\|^2\right] = \mathbb{E}\left[\left\|z^{(\ell')}\right\|^2\right] \quad \text{for all } \ell, \ell', \tag{3}$$

where $\mathbb{E}\left[\cdot\right]$ denotes the average over initialization. This condition essentially corresponds to asking for equal scales in any hidden layer representations. Our first result, which is closely related to the ideas developed for MLPs (Theorem 2 in (Defazio & Bottou, 2022)), gives a simple criterion for ensuring that equation 3 holds for a DAG-structured neural network with ReLU layers:

**Initialization scaling for DAG.** For a neural network with directed acyclic computational graph $\mathcal{G} = (V, E)$, $V = [0, L+1]$, width $n$, and ReLU activations, the information flow condition equation 3 is satisfied on average over a centred random input $x$ if

$$C^{(\ell', \ell)} = \frac{2}{d_{\text{in}}^{(\ell')}}, \tag{4}$$

where for each vertex $\ell \in [0, L+1]$, we've written $d_{\text{in}}^{(\ell)}$ for its in-degree in $\mathcal{G}$. This implies that, when we initialize a layer, we should consider how many features flow into it. We include our derivations in Appendix A.

### 3.3 TOPOLOGY-AWARE LEARNING RATES

The purpose of this section is to describe a specific method for choosing learning rates for DAG networks. Our approach follows the maximal update ($\mu$P) heuristic proposed in (Yang et al., 2022). Specifically, given a DAG network recall from equation 1 that

$$z^{(\ell)}(x) = \left(z_i^{(\ell)}(x), i = 1, \ldots, n\right)$$

denotes the pre-activations at layer $\ell$ corresponding to a network input $x$. The key idea in (Yang et al., 2022) is to consider the change $\Delta z_i^{(\ell)}$ of $z_i^{(\ell)}$ from one gradient descent step on training loss $\mathcal{L}$:

$$\Delta z_i^{(\ell)} = \sum_{\mu \leq \ell} \partial_\mu z_i^{(\ell)} \Delta \mu = -\eta \sum_{\mu \leq \ell} \partial_\mu z_i^{(\ell)} \partial_\mu \mathcal{L}, \tag{5}$$

where $\mu$ denotes a network weight, $\mu \leq \ell$ means that $\mu$ is a weight in $W^{(\ell', \ell)}$ for some $\ell' \leq \ell - 1$, $\Delta \mu$ for the change in $\mu$ after one step of GD, $\eta$ is the learning rate. By symmetry, all weights in layer $\ell$ should have the same learning rate at initialization and we will take as an objective function the MSE loss:

$$\mathcal{L}(\theta) = \frac{1}{|\mathcal{B}|} \sum_{(x,y) \in \mathcal{B}} \frac{1}{2} \left\| z^{(L+1)}(x; \theta) - y \right\|^2 \tag{6}$$

over a batch $\mathcal{B}$ of training datapoints. The $\mu$P heuristic posited in (Yang et al., 2022) states that the best learning rate is the largest one for which $\Delta z_i^{(\ell)}$ remains $O(1)$ for all $\ell = 1, \ldots, L+1$ independent of the network width $n$. To make this precise, we will mainly focus on analyzing the average squared change (second moment) of $\Delta z_i^{(\ell)}$ and ask to find learning rates such that

$$\mathbb{E}\left[\left(\Delta z_i^{(\ell)}\right)^2\right] = \Theta(1) \quad \text{for all } \ell \in [1, L+1] \tag{7}$$

with mean-field initialization, which coincides with equation 2 except that the final layer weights are initialized with variance $1/n^2$ instead of $1/n$. Our next result provides a way to estimate how the $\mu$P learning rate (i.e. the one determined by equation 7) depends on the graph topology. To state it, we need two definitions:

- **DAG Width.** Given a DAG $\mathcal{G} = (V, E)$ with $V = \{0, \ldots, L+1\}$ we define its width $P$ to be the number of unique directed paths from the input vertex 0 to output vertex $L+1$.

- **Path Depth.** Given a DAG $\mathcal{G} = (V, E)$, the number of ReLU activations in a directed path $p$ from the input vertex 0 to the output vertex $L+1$ is called its depth, denoted as $L_p$ ($p = 1, \cdots, P$).

We are now ready to state the main result of this section:

**Learning rate scaling in DAG.**   Consider a ReLU network with associated DAG $\mathcal{G} = (V, E)$, hidden layer width $n$, hidden layer weights initialized as in § 3.2 and the output weights initialized with mean-field scaling. Consider a batch with one training datapoint $(x, y) \in \mathbb{R}^{2n}$ sampled independently of network weights and satisfying

$$\mathbb{E}\left[\frac{1}{n}\|x\|^2\right] = 1, \quad \mathbb{E}\left[y\right] = 0, \ \mathrm{Var}[y] = 1.$$

The $\mu$P learning rate ($\eta^*$) for hidden layer weights is independent of $n$ but has a non-trivial dependence of depth, scaling as

$$\eta^* \simeq c \cdot \left(\sum_{p=1}^{P} L_p^3\right)^{-1/2}, \tag{8}$$

where the sum is over $P$ unique directed paths through $\mathcal{G}$, and the implicit constant $c$ is independent of the graph topology and of the hidden layer width $n$.

This result indicates that, we should use smaller learning rates for networks with large depth and width (of its graph topology); and vice versa. We include our derivations in Appendix B.

## 3.4    Learning Rates in DAG Network with Convolutional Layers

In this section, we further expand our scaling principles to convolutional neural networks, again, with arbitrary graph topologies as we discussed in § 3.1.

**Setting.**   Let $x \in \mathbb{R}^{n \times m}$ be the input, where $n$ is the number of input channels and $m$ is the number of pixels. Given $z^{(\ell)} \in \mathbb{R}^{n \times m}$ for $\ell \in [1, L+1]$, we first use an operator $\phi(\cdot)$ to divide $z^{(\ell)}$ into $m$ patches. Each patch has size $qn$ and this implies a mapping of $\phi(z^{(\ell)}) \in \mathbb{R}^{qn \times m}$. For example, when the stride is 1 and $q = 3$, we have:

$$\phi(z^{(\ell)}) = \begin{pmatrix} \left(z_{1,0:2}^{(\ell)}\right)^\top, & \cdots & , \left(z_{1,m-1:m+1}^{(\ell)}\right)^\top \\ \cdots, & \cdots, & \cdots \\ \left(z_{n,0:2}^{(\ell)}\right)^\top, & \cdots, & \left(z_{n,m-1:m+1}^{(\ell)}\right)^\top \end{pmatrix},$$

where we let $z_{:,0}^{(\ell)} = z_{:,m+1}^{(\ell)} = 0$, i.e., zero-padding. Let $W^{(\ell)} \in \mathbb{R}^{n \times qn}$. we have

$$z^{(\ell)}(x) = \begin{cases} x, & \ell = 0 \\ \sum_{(\ell',\ell) \in E} W^{(\ell',\ell)} \sigma(\phi(z^{(\ell')}(x))), & \ell = 1, \ldots, L+1 \end{cases}.$$

**Learning rate scaling in CNNs.**   Consider a ReLU-CNN network with associated DAG $\mathcal{G} = (V, E)$, kernel size $q$, hidden layer width $n$, hidden layer weights initialized as in § 3.2, and the output weights initialized with mean-field scaling. The $\mu$P learning rate $\eta^*$ for hidden layer weights is independent of $n$ but has a non-trivial dependence of depth, scaling as

$$\eta^* \simeq c \cdot \left(\sum_{p=1}^{P} L_p^3\right)^{-1/2} \cdot q^{-1}. \tag{9}$$

This result implies that in addition to network depth and width, CNNs with a larger kernel size should use a smaller learning rate. We include our derivations in Appendix C.

## 3.5    Implications on Architecture Design

We would like to emphasize that, beyond training deep networks to better performance, our work will imply a much broader impact in questioning the credibility of benchmarks and algorithms for automated machine learning (**AutoML**).

Designing novel networks is vital to the development of deep learning and artificial intelligence. Specifically, Neural Architecture Search (NAS) dramatically speeds up the discovery of novel architectures in a principled way (Zoph & Le, 2016; Pham et al., 2018; Liu et al., 2018), and

meanwhile many of them archive state-of-the-art performance. The validation of NAS algorithms largely hinges on the quality of architecture benchmarks. However, in most literature, each NAS benchmark adapts the same set of training protocols when comparing diverse neural architectures, largely due to the prohibitive cost of searching the optimal hyperparameter for each architecture.

To our best knowledge, there exist very few in-depth studies of "how to train" architectures sampled from the NAS search space, thus we are motivated to examine the unexplored "hidden gem" question: *how big a difference can be made if we specifically focus on improving the training hyperparameters?* We will answer this question in § 4.3.

## 4 EXPERIMENTS

In our experiments, we will apply our learning rate scaling to weights and bias. We study our principles on MLPs, CNNs, and networks with advanced architectures from NAS-Bench-201 (Dong & Yang, 2020). All our experiments are repeated for three random runs. We also include more results in the Appendix (§ E.1 for ImageNet, § E.2 for the GeLU activation).

**We adapt our principles as follows:**

1. Find the base maximal learning rate of the base architecture with the smallest depth ($L = 1$ in our case: "input→hidden→output"): conduct a grid search over a range of learning rates, and find the maximal learning rate which achieves the smallest training loss at the end of one epoch[1]

2. Initialize the target network (of deeper layers or different topologies) according to equation 4.

3. Train the target network by scaling the learning rate based on equation 8 for MLPs or equation 9 for CNNs.

The above steps are much cheaper than directly tuning hyperparameters for heavy networks, since the base network is much smaller. We verify this principle for MLPs with different depths in Appendix D.

### 4.1 MLPS WITH TOPOLOGY SCALING

We study MLP networks by scaling the learning rate to different graph structures based on equation 8. In addition to the learning rate, variances of initializations are also adjusted based on the in-degree of each layer in a network.

To verify our scaling rule of both learning rates and initializations, we first find the "ground truth" maximal learning rates via grid search on MLPs with different graph structures, and then compare them with our scaled learning rates. As shown in Figure 2, our estimation strongly correlates with the "ground truth" maximal learning rates ($r = 0.838$). This result demonstrates that our learning rate scaling principle can be generalized to complicated network architectures.

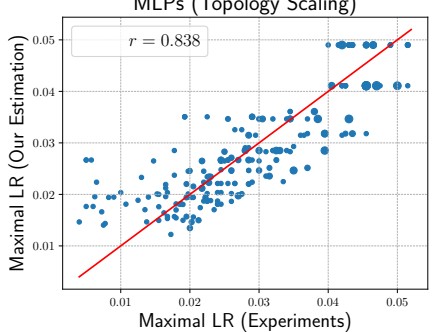

Figure 2: MLPs with different topologies (graph structures). X-axis: "ground truth" maximal learning rates found by grid search. Y-axis: estimated learning rates by our principle (equation 8). The red line indicates the identity. Based on the "ground truth" maximal learning rate of the basic MLP with $L = 1$, we scale up both learning rates and initialization to diverse architectures. The radius of a dot indicates the variance over three random runs. Data: CIFAR-10.

### 4.2 CNNS WITH TOPOLOGY SCALING

We further extend our study to CNNs. Our base CNN is of $L = 1$ and kernel size $q = 3$, and we scale the learning rate to CNNs of both different graph structures and kernel sizes, based on equation 9. Both the learning rate and the variances of initializations are adjusted for each network.

---

[1] Rationales for finding maximal learning rates for the 1st epoch: 1) Finding hyperparameters in early training is both practically effective and widely adopted (Tab. 1 and 4 of (Bohdal et al., 2022), Fig. 2 and 3a of (Egele et al., 2023). Analyzing subsequent training epochs will require more computations and careful design of HPO algorithms. 2) $\mu$P Yang et al. (2022) finds optimal learning rates in early training (Fig. 4 and 19). Exploiting more training epochs (for HPO purposes) is orthogonal to our method and is beyond the scope of our work.

Again, we first collect the "ground truth" maximal learning rates by grid search on CNNs with different graph structures and kernel sizes, and compare them with our estimated ones. As shown in Figure 3, our estimation strongly correlates with the "ground truth" maximal learning rates of heterogeneous CNNs ($r = 0.856$). This result demonstrates that our initialization and learning rate scaling principle can be further generalized to CNNs with sophisticated network architectures and different kernel sizes.

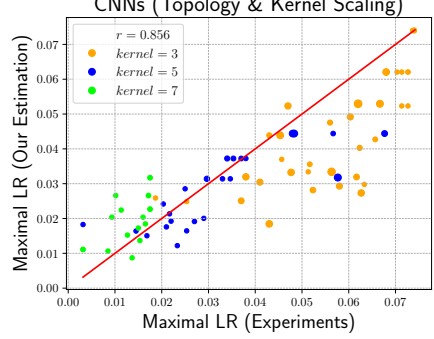

Figure 3: CNNs with different graph structures and kernel sizes. The x-axis shows the "ground truth" maximal learning rates found by grid search. The y-axis shows the estimated learning rates by our principle (equation 9). The red line indicates the identity. Based on the "ground truth" maximal learning rate of the CNN with $L = 1$ and kernel size as 3, we scale up both learning rates and initialization to diverse architectures. The radius of a dot indicates the variance over three random runs. Data: CIFAR-10.

### 4.3 BREAKING NETWORK RANKINGS IN NAS

Networks' performance ranking is vital to the comparison of NAS algorithms (Guo et al., 2020; Mellor et al., 2021; Chu et al., 2021; Chen et al., 2021). If the ranking cannot faithfully reflect the true performance of networks, we cannot trust the quality of architecture benchmarks. Here we will show that: by simply better train networks, we can easily improve accuracies and break rankings of architectures (those in benchmarks or searched by AutoML algorithms).

We will use NAS-Bench-201 (Dong & Yang, 2020) as the test bed. NAS-Bench-201 provides a cell-based search space NAS benchmark, the network's accuracy is directly available by querying the database, benefiting the study of NAS methods without network evaluation. It contains five heterogeneous operator types: *none* (*zero*), *skip connection*, *conv*$1 \times 1$, *conv*$3 \times 3$ *convolution*, and *average pooling*$3 \times 3$. Accuracies are provided for three datasets: CIFAR-10, CIFAR-100, ImageNet16-120. However, due to the prohibitive cost of finding the optimal hyperparameter set for each architecture, the performance of all 15,625 architectures in NAS-Bench-201 is obtained with the same protocol (learning rate, initialization, etc.). Sub-optimal hyperparameters may setback the maximally achievable performance of each architecture, and may lead to unreliable comparison between NAS methods.

We randomly sample architectures from NAS-Bench-201, and adopt the same training protocol as Dong & Yang (2020) (batch size, warm-up, learning rate decay, etc.). The only difference is that we use our principle (§ 3.4) to re-scale learning rates and initializations for each architecture, and train networks till converge. Results are shown in Figure 4. We derive three interesting findings from our plots to contribute to the NAS community from novel perspectives:

1. Compared with the default training protocol on NAS-Bench-201, neural architectures can benefit from our scaled learning rates and initialization with much better performance improvements (left column in Figure 4). This is also a step further towards an improved "gold standard" NAS benchmark. Specifically, on average we outperform the test accuracy in NAS-Bench-201 by 0.44 on CIFAR-10, 1.24 on CIFAR-100, and 2.09 on ImageNet16-120. We also show our superior performance over $\mu$P in Appendix E.3.

2. Most importantly, compared with the default training protocol on NAS-Bench-201, **architectures exhibit largely different performance rankings, especially for top-performing ones** (middle column in Figure 4). This implies: with every architecture better-trained, state-of-the-art NAS algorithms evaluated on benchmarks may also be re-ranked. This is because: many NAS methods (e.g. works summarized in (Dong et al., 2021)) are agnostic to hyperparameters (i.e. search methods do not condition on training hyperparameters). Changes in the benchmark's training settings will not affect architectures they searched, but instead will lead to different training performances and rankings of searched architectures.

3. By adopting our scaling method, the performance gap between architectures is mitigated (right column in Figure 4). In other words, our method enables all networks to be converged similarly well, thus less distinguishable in performance.

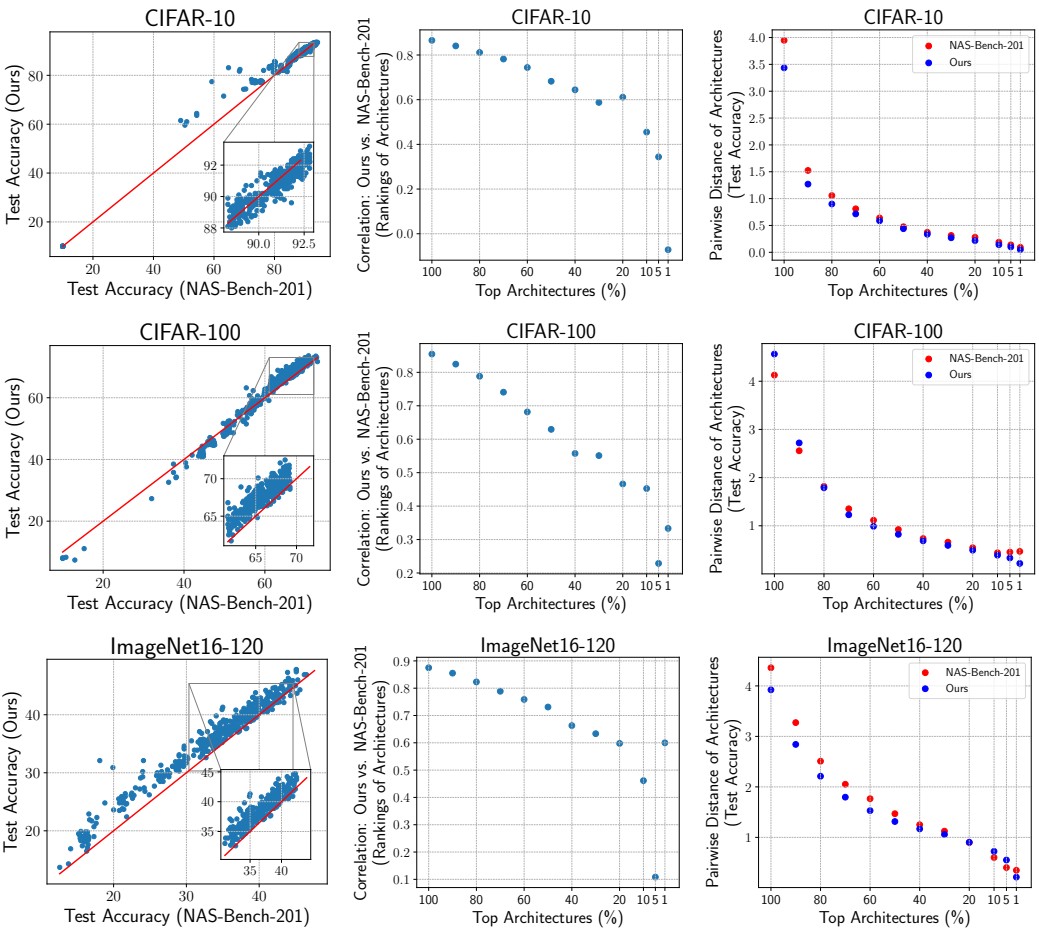

Figure 4: We adopt our scaling principles to existing architecture benchmarks. **Left column**: better accuracy. For different network architectures, we plot the accuracy trained with different scaling principles (y-axis "Re-scaled") against the accuracy trained with a fixed training recipe (x-axis "NAS-Bench-201"). Each dot represents a unique architecture with different layer types and topologies (Figure 1 in (Dong & Yang, 2020)). Our principle (blue dots) achieves better accuracy compared to the benchmark (red line $y = x$). **Middle column**: network rankings in benchmarks are fragile. We compare networks' performance rankings at different top $K\%$ percentiles ($K = 100, 90, \cdots, 10, 5, 1$; bottom right dots represent networks on the top-right in the left column), trained by our method vs. benchmarks, and find better networks are ranked more differently from the benchmark. This indicates current network rankings in benchmarks (widely used to compare NAS algorithms) can be easily broken by simply better train networks. **Right column**: less distinguishable architectures. We plot the pairwise performance gaps between different architectures, and find our principle makes networks similar and less distinguishable in terms of their accuracies.

## 5 CONCLUSION

Training a high-quality deep neural network requires choosing suitable hyperparameters, which are non-trivial and expensive. However, most scaling or optimization methods are agnostic to the choice of networks, and thus largely ignore the impact of neural architectures on hyperparameters. In this work, by analyzing the dependence of pre-activations on network architectures, we propose a scaling principle for both the learning rate and initialization that can generalize across MLPs and CNNs, with different depths, connectivity patterns, and kernel sizes. Comprehensive experiments verify our principles. More importantly, our strategy further sheds light on potential improvements in current benchmarks for architecture design. We point out that by appropriately adjusting the learning rate and initializations per network, current rankings in these benchmarks can be easily broken.Our work contributes to both network training and model design for the Machine Learning community. Limitations of our work: 1) Principle of width-dependent scaling of the learning rate. 2) Characterization of depth-dependence of learning rates beyond the first step / early training of networks. 3) Depth-dependence of learning rates with the existence of normalization layers.

ACKNOWLEDGMENTS

B. Hanin and Z. Wang are supported by NSF Scale-MoDL (award numbers: 2133806, 2133861).

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

## A    ARCHITECTURE-AWARE INITIALIZATION

*Derivations for § 3.2.* Consider a random network input $x$ sampled from a distribution for which $x$ and $-x$ have the same distribution. Let us first check that the joint distribution of the pre-activation vectors $z^{(\ell')}(x)$ with $\ell' = 0, \ldots, L + 1$ is symmetric around 0. We proceed by induction on $\ell$. When $\ell = 0$ this statement is true by assumption. Next, suppose we have proved the statement for $0, \ldots, \ell$. Then, since $z^{(\ell+1)}(x)$ is a linear function of $z^{(0)}(x), \ldots, z^{(\ell)}(x)$, we conclude that it is also symmetric around 0. A direct consequence of this statement is that, by symmetry,

$$\mathbb{E}\left[\sigma^2\left(z_j^{(\ell)}(x)\right)\right] = \frac{1}{2}\mathbb{E}\left[\left(z_j^{(\ell)}(x)\right)^2\right], \qquad \forall j = 1, \ldots, n, \ \ell = 0, \ldots, L + 1 \qquad (10)$$

Let us fix $k = 1, \ldots, n$ and a vertex $\ell$ we calculate $C_W$ in equation 2:

$$\begin{aligned}
\mathbb{E}\left[\left\|z^{(\ell)}\right\|^2\right] &= \mathbb{E}\left[\sum_{j=1}^{n}\left(z_j^{(\ell)}\right)^2\right] \\
&= \mathbb{E}\left[\sum_{j=1}^{n}\left(\sum_{(\ell',\ell)\in E}\sum_{j'=1}^{n} W_{jj'}^{(\ell',\ell)}\sigma\left(z_{j'}^{(\ell')}\right)\right)^2\right] \\
&= \sum_{j=1}^{n}\sum_{(\ell',\ell)\in E}\sum_{j'=1}^{n}\frac{C_W^{(\ell',\ell)}}{n}\frac{1}{2}\mathbb{E}\left[\left(z_{j'}^{(\ell')}\right)^2\right] \\
&= \sum_{(\ell,\ell')\in E}\frac{1}{2}C_W^{(\ell,\ell')}\mathbb{E}\left[\left\|z^{(\ell')}\right\|^2\right].
\end{aligned} \qquad (11)$$

We ask $\mathbb{E}\left[\left\|z^{(\ell)}\right\|^2\right] = \mathbb{E}\left[\left\|z^{(\ell')}\right\|^2\right]$, which yields

$$C_W^{(\ell',\ell)} = \frac{2}{d_{\text{in}}^{(\ell)}}. \qquad (12)$$

We now seek to show

$$\mathbb{E}\left[\left\|z^{(\ell)}\right\|^2\right] = \mathbb{E}\left[\left\|z^{(\ell')}\right\|^2\right] \qquad \forall \ell, \ell' \in V. \qquad (13)$$

To do so, let us define for each $\ell = 0, \ldots, L + 1$

$$d(\ell, L + 1) := \{\text{length of longest directed path in } \mathcal{G} \text{ from } \ell \text{ to } L + 1\}.$$

The relation equation 13 now follows from a simple argument by induction show that by induction on $d(\ell, L+1)$, starting with $d(\ell, L+1) = 0$. Indeed, the case $d(\ell, L+1) = 0$ simply corresponds to $\ell = L+1$. Next, note that if $d(\ell, L+1) > 0$ and if $(\ell', \ell) \in E$, then $d(\ell, L+1) < d(\ell', L+1)$ by the maximality of the path length in the definition of $d(\ell, L+1)$. Hence, substituting equation 12 into equation 11 completes the proof of the inductive step.

$\square$

## B  ARCHITECTURE-DEPENDENT LEARNING RATES (FOR § 3.3)

We start by setting some notation. Specifically, we fix a network input $x \in \mathbb{R}^{n_0}$ at which we study both the forward and backward pass. We denote by

$$z_i^{(\ell)} := z_i^{(\ell)}(x), \qquad z^{(\ell)} := z^{(\ell)}(x)$$

the corresponding pre-activations at vertex $\ell$. We assume our network has a uniform width over layers ($n_\ell \equiv n$), and parameter-dependently learning rates:

$$\eta_\mu = \text{ learning rate of } \mu.$$

We will restrict to the case $\eta_\mu = \eta$ at the end. Further, recall from equation 6 that we consider the empirical MSE over a batch $\mathcal{B}$ with a single element $(x, y)$ and hence our loss is

$$\mathcal{L} = \frac{1}{2}\left(z^{(L+1)} - y\right)^2.$$

The starting point for derivations in § 3.3 is the following Lemma

**Lemma B.1** (Adapted from Lemma 2.1 in Jelassi et al. (2023)). *For $\ell = 1, \ldots, L$, we have*

$$\mathbb{E}\left[(\Delta z_i^{(\ell)})^2]\right] = A^{(\ell)} + B^{(\ell)},$$

*where*

$$A^{(\ell)} := \mathbb{E}\left[\frac{1}{n^2} \sum_{\mu_1, \mu_2 \leq \ell} \eta_{\mu_1} \eta_{\mu_2} \partial_{\mu_1} z_1^{(\ell)} \partial_{\mu_2} z_1^{(\ell)}\right. \tag{14}$$

$$\times \frac{1}{\left(d_{in}^{(L+1)}\right)^2} \sum_{(\ell_1', L+1), (\ell_2', L+1) \in E} \frac{1}{n^2} \tag{15}$$

$$\left.\times \sum_{j_1, j_2 = 1}^{n} \left\{\partial_{\mu_1} z_{j_1}^{(\ell_1')} \partial_{\mu_2} z_{j_1}^{(\ell_1')} \left(z_{j_2}^{(\ell_2')}\right)^2 + 2 z_{j_1}^{(\ell_1')} \partial_{\mu_1} z_{j_1}^{(\ell_2')} z_{j_2}^{(\ell_1')} \partial_{\mu_2} z_{j_2}^{(\ell_2')}\right\}\right],$$

$$B^{(\ell)} := \mathbb{E}\left[\frac{1}{n} \sum_{\mu_1, \mu_2 \leq \ell} \eta_{\mu_1} \eta_{\mu_2} \partial_{\mu_1} z_1^{(\ell)} \partial_{\mu_2} z^{(\ell)} \frac{1}{d_{in}^{(L+1)}} \sum_{(\ell', L+1) \in E} \frac{1}{n} \sum_{j=1}^{n} \partial_{\mu_1} z_j^{(\ell')} \partial_{\mu_2} z_j^{(\ell')}\right]. \tag{16}$$

*Proof.* The proof of this result follows very closely the derivation of Lemma 2.1 in Jelassi et al. (2023) which considered only the case of MLPs. We first expand $\Delta z_i^{(\ell)}$ by applying the chain rule:

$$\Delta z_i^{(\ell)} = \sum_{\mu \leq \ell} \partial_\mu z_i^{(\ell)} \Delta \mu, \tag{17}$$

where the sum is over weights $\mu$ that belong to some weight matrix $W^{(\ell'', \ell')}$ for which there is a directed path from $\ell'$ to $\ell$ in $\mathcal{G}$ and we've denoted by $\Delta\mu$ the change in $\mu$ after one step of GD. The SGD update satisfies:

$$\Delta\mu = -\frac{\eta_\mu}{2} \partial_\mu \left(z^{(L+1)} - y\right)^2 = -\eta_\mu \partial_\mu z^{(L+1)} \left(z^{(L+1)} - y\right), \tag{18}$$

where we've denoted by $(x, y)$ the training datapoint in the first batch. We now combine equation 17 and equation 18 to obtain:

$$\Delta z_i^{(\ell)} = \sum_{\mu \leq \ell} \eta_\mu \partial_\mu z_i^{(\ell)} \partial_\mu z^{(L+1)} \left( y - z^{(L+1)} \right). \tag{19}$$

Using equation 19, we obtain

$$
\mathbb{E}\left[\left(\Delta z_i^{(\ell)}\right)^2\right] = \mathbb{E}\left[\left(\sum_{\mu \leq \ell} \eta_\mu \partial_\mu z_1^{(\ell)} \partial_\mu z_1^{(L+1)} \left(z^{(L+1)} - y\right)\right)^2\right]
$$

$$
= \mathbb{E}\left[\sum_{\mu_1, \mu_2 \leq \ell} \eta_{\mu_1} \eta_{\mu_2} \partial_{\mu_1} z_1^{(\ell)} \partial_{\mu_2} z_1^{(\ell)} \partial_{\mu_1} z^{(L+1)} \partial_{\mu_2} z^{(L+1)} \mathbb{E}_y\left[\left(z^{(L+1)} - y\right)^2\right]\right].
$$
$$\tag{20}$$

Here, we've used that by symmetry the answer is independent of $i$. Taking into account the distribution of $z^{(L+1)}$ and $y$, we have

$$\mathbb{E}_y\left[\left(z^{(L+1)} - y\right)^2\right] = (z^{(L+1)})^2 + 1 \tag{21}$$

We plug equation 21 in equation 20 and obtain

$$\mathbb{E}[(\Delta z_i^{(\ell)})^2] = A^{(\ell)} + B^{(\ell)}, \tag{22}$$

where

$$A^{(\ell)} = \mathbb{E}\left[\sum_{\mu_1, \mu_2 \leq \ell} \eta_{\mu_1} \eta_{\mu_2} \partial_{\mu_1} z^{(\ell)} \partial_{\mu_2} z^{(\ell)} \partial_{\mu_1} z^{(L+1)} \partial_{\mu_2} z^{(L+1)} \left(z^{(L+1)}\right)^2\right] \tag{23}$$

$$B^{(\ell)} = \mathbb{E}\left[\sum_{\mu_1, \mu_2 \leq \ell} \eta_{\mu_1} \eta_{\mu_2} \partial_{\mu_1} z_1^{(\ell)} \partial_{\mu_2} z^{(\ell)} \partial_{\mu_1} z^{(L+1)} \partial_{\mu_2} z^{(L+1)}\right]. \tag{24}$$

By definition, we have

$$z^{(L+1)} = \sum_{(\ell', L+1) \in E} \sum_{j=1}^n W_j^{(\ell', L+1)} \sigma\left(z_j^{(\ell')}\right), \qquad W_j^{(\ell', L+1)} \sim \mathcal{N}\left(0, \frac{1}{n^2}\right).$$

Therefore, integrating out weights of the form $W^{(\ell', L+1)}$ in equation 24 yields

$$
B^{(\ell)} = \mathbb{E}\left[\frac{1}{n} \sum_{\mu_1, \mu_2 \leq \ell} \eta_{\mu_1} \eta_{\mu_2} \partial_{\mu_1} z^{(\ell)} \partial_{\mu_2} z^{(\ell)} \frac{1}{d_{\text{in}}^{(L+1)}} \sum_{(\ell', L+1) \in E} \frac{2}{n} \sum_{j=1}^n \partial_{\mu_1} \sigma\left(z_j^{(\ell')}\right) \partial_{\mu_2} \sigma\left(z_j^{(\ell')}\right)\right]
$$

$$
= \mathbb{E}\left[\frac{1}{n} \sum_{\mu_1, \mu_2 \leq \ell} \eta_{\mu_1} \eta_{\mu_2} \partial_{\mu_1} z^{(\ell)} \partial_{\mu_2} z^{(\ell)} \frac{1}{d_{\text{in}}^{(L+1)}} \sum_{(\ell', L+1) \in E} \frac{1}{n} \sum_{j=1}^n \partial_{\mu_1} z_j^{(\ell')} \partial_{\mu_2} z_j^{(\ell')}\right],
$$

where in the last equality we used equation 10. This yields the desired formula for $B^{(\ell)}$. A similar computation yields the expression for $A^{(\ell)}$. ☐

As in the proof of Theorem 1.1 in Jelassi et al. (2023), we have

$$A^{(\ell)} = O(n^{-1})$$

due to the presence of an extra pre-factor of $1/n$. Our derivation in § 3.3 therefore comes down to finding how $B^{(\ell)}$ is influenced by the topology of the graph $\mathcal{G}$.

We then re-write the high-level change of pre-activations (equation 5) in an arbitrary graph topology. We need to accumulate all changes of pre-activations that flow into a layer (summation over in-degrees $\ell' \to L + 1$):

$$\Delta z_i^{(L+1)} = \sum_{\ell' \to L+1} \sum_{\mu \leq \ell'} \partial_\mu z_i^{(\ell')} \Delta\mu \tag{25}$$

Therefore, for the change of pre-activation of layer $L + 1$, we can simplify the analysis to each individual layer $\ell'$ that connects $L + 1$ (i.e. edge $(\ell' \to L + 1) \in E$), and then sum up all layers that flow into layer $\ell$.

Next, we focus on deriving $B^{(\ell)}$ of each individual path in the case of the DAG structure of network architectures. It is important that here we adopt our architecture-aware initialization (§ 3.2).

$$
\begin{aligned}
B^{(\ell)} &= \mathbb{E}\left[ \sum_{\mu_1,\mu_2 \leq \ell} \eta_{\mu_1}\eta_{\mu_2} \partial_{\mu_1} z_i^{(\ell)} \partial_{\mu_2} z_i^{(\ell)} \sum_{(\ell',L+1)\in E} \frac{1}{d_{\text{in}}^{(L+1)}} \sum_{j=1}^{n} \partial_{\mu_1} z_j^{(\ell')} \partial_{\mu_2} z_j^{(\ell')} \right] \\
&= \mathbb{E}\left[ \sum_{\mu_1,\mu_2 \leq \ell} \eta_{\mu_1}\eta_{\mu_2} \partial_{\mu_1} z_i^{(\ell)} \partial_{\mu_2} z_i^{(\ell)} \partial_{\mu_1} z_j^{(\ell')} \partial_{\mu_2} z_j^{(\ell')} \right] \\
&= \cdots \\
&= \mathbb{E}\left[ \sum_{\mu_1,\mu_2 \leq \ell} \eta_{\mu_1}\eta_{\mu_2} \partial_{\mu_1} z_i^{(\ell)} \partial_{\mu_2} z_i^{(\ell)} \partial_{\mu_1} z_j^{(\ell)} \partial_{\mu_2} z_j^{(\ell)} \right].
\end{aligned}
\tag{26}
$$

Thus, we can see that with our architecture-aware initialization, $B^{(\ell)}$ reduce back to the basic sequential MLP case in Theorem 1.1 in Jelassi et al. (2023).

Therefore, we have:

$$B^{(L+1)} \simeq \sum_{p=1}^{P} \Theta(\eta^2 L_p^3).$$

Thus

$$\eta \simeq \left( \sum_{p=1}^{P} L_p^3 \right)^{-1/2}.$$

where $P$ is the total number of end-to-end paths that flow from the input to the final output $z^{L+1}$, and $L_p$ is the number of ReLU layers on each end-to-end path.

## C    LEARNING RATE SCALING FOR CNNS

*Derivations for § 3.4.* In addition to $B^{(\ell)}$ in equation 24, we further refer to the contributing term of $B^{(\ell)}$ in Jelassi et al. (2023) when $\mu_1 \leq \ell - 1$ and $\mu_2 \in \ell$ (or vice versa), denoted as $C^{(\ell)}$:

$$C^{(\ell)} := \mathbb{E}\left[ \frac{1}{n} \sum_{\mu \leq \ell} \eta_\mu \frac{1}{n^2} \sum_{j_1,j_2=1}^{n} \left( z_{j_1}^{(\ell)} \partial_\mu z_{j_2}^{(\ell)} \right)^2 \right]. \tag{27}$$

We start from deriving the recursion of $B^{(\ell)}$ of each individual end-to-end path for convolutional layers of a kernel size as $q$. Again, we assume our network has a uniform width over layers ($n_\ell \equiv n$).

If $\mu_1, \mu_2 \in \ell$ then the contribution to equation 24 is

$$\left( \eta^{(\ell)} \right)^2 q^2 \mathbb{E}\left[ \frac{1}{n^2} \sum_{j_1,j_2=1}^{n} \left( \sigma_{j_1}^{(\ell-1)} \sigma_{j_2}^{(\ell-1)} \right)^2 \right] = \left( \eta^{(\ell)} \right)^2 q^2 \frac{4}{n^2} \|x\|^2 e^{5 \sum_{\ell'=1}^{\ell-2} \frac{1}{n}}$$

When $\mu_1 \leq \ell - 1$ and $\mu_2 \in \ell$ (or vice versa) the contribution to equation 24 is

$$2\eta^{(\ell)} q \mathbb{E}\left[ \frac{1}{n} \sum_{\mu_1 \leq \ell-1} \eta_{\mu_1} \frac{1}{n} \sum_{k=1}^{n} \left( \sigma_k^{(\ell-1)} \right)^2 \frac{1}{n} \sum_{j=1}^{n} \left( \partial_{\mu_1} z_j^{(\ell)} \right)^2 \right] = \eta^{(\ell)} q C^{(\ell-1)}.$$

Finally, when $\mu_1, \mu_2 \leq \ell - 1$ we find the contribution to equation 24 becomes

$$\mathbb{E}\left[\frac{1}{n}\sum_{\mu_1,\mu_2 \leq \ell-1} \eta_{\mu_1}\eta_{\mu_2}\left\{\frac{1}{n}\left(\partial_{\mu_1}z_1^{(\ell)}\partial_{\mu_2}z_1^{(\ell)}\right)^2 + \left(1 - \frac{1}{n}\right)\partial_{\mu_1}z_1^{(\ell)}\partial_{\mu_2}z_1^{(\ell)}\partial_{\mu_1}z_2^{(\ell)}\partial_{\mu_2}z_2^{(\ell)}\right\}\right]$$
$$= \left(1 + \frac{1}{n}\right)B^{(\ell-1)} + \frac{1}{n}\widetilde{B}^{(\ell-1)}.$$

Therefore, we have

$$B^{(\ell)} = \left(\eta^{(\ell)}\right)^2 q^2 \frac{4}{n^2}\|x\|^2 e^{5\sum_{\ell'=1}^{\ell-2}\frac{1}{n}} + \eta^{(\ell)}qC^{(\ell-1)} + \left(1 + \frac{1}{n}\right)B^{(\ell-1)} + \frac{1}{n}\widetilde{B}^{(\ell-1)}$$

$$\frac{1}{n}\widetilde{B}^{(\ell)} = \left(\eta^{(\ell)}\right)^2 q^2 \frac{4\|x\|^4}{n^2}e^{5\sum_{\ell'=1}^{\ell-2}\frac{1}{n}} + \eta^{(\ell)}qC^{(\ell-1)} + \frac{1}{n}\widetilde{B}^{(\ell-1)} + \frac{2}{n^2}B^{(\ell-1)}$$

We further derive $C^{(\ell)}$. When $\mu \in \ell$ the contribution to equation 27 is

$$\eta^{(\ell)}q\mathbb{E}\left[\frac{1}{n^2}\sum_{j_1,j_2=1}^{n}\left(z_{j_1}^{(\ell-1)}z_{j_2}^{(\ell-1)}\right)^2\right] = \eta^{(\ell)}q\frac{\|x\|^4}{n^2}e^{5\sum_{\ell'=1}^{\ell-1}\frac{1}{n}}.$$

When $\mu \leq \ell - 1$ the contribution to equation 27 is

$$\frac{1}{n}\mathbb{E}\left[\sum_{\mu \leq \ell-1}\eta_\mu\left\{\frac{1}{n}\left(\partial_\mu z_1^{(\ell)}z_1^{(\ell)}\right)^2 + \left(1 - \frac{1}{n}\right)\left(\partial_\mu z_1^{(\ell)}\right)^2\left(z_2^{(\ell)}\right)^2\right\}\right]$$
$$= C^{(\ell-1)} + \frac{1}{n}\widetilde{C}^{(\ell-1)}.$$

Therefore,

$$C^{(\ell)} = \frac{1}{2}\eta^{(\ell)}q\frac{\|x\|^4}{n^2}e^{5\sum_{\ell'=1}^{\ell-1}\frac{1}{n}} + \frac{1}{n}C^{(\ell-1)} + \left(1 + \frac{1}{n}\right)\widetilde{C}^{(\ell-1)}$$

Finally, for each end-to-end path, we have

$$B^{(\ell)} \simeq \Theta(\eta^2\ell^3 q^2).$$

Therefore, together with § 3.3, we want

$$\eta \simeq \left(\sum_{p=1}^{P}L_p^3\right)^{-1/2}\cdot q^{-1}.$$

$\square$

## D    MLPs WITH DEPTH SCALING

We verify the depth-wise scaling rule in Jelassi et al. (2023). We scale a vanilla feedforward network by increasing its depth (adding more Linear-ReLU layers). Starting from the most basic feedforward network with $L = 3$ (an input layer, a hidden layer, plus an output layer), we first scan a wide range of learning rates and find the maximal learning rate. We then scale the learning rate to feedforward networks of different depths according to equation 8. Different feedforward networks share the same initialization since both the out-degree and in-degree of all layers are 1.

To verify the scaling results, we also conduct the grid search of learning rates for feedforward networks of different depths, and compare them with the scaled learning rates. As shown in Figure 5, on CIFAR-10 the estimation strongly correlates with the "ground truth" maximal learning rates ($r = 0.962$), and the plotted dots are very close to the identity line. This result demonstrates that the depth-wise learning rate scaling principle in Jelassi et al. (2023) is highly accurate across feedforward neural networks of different depths and across different datasets.

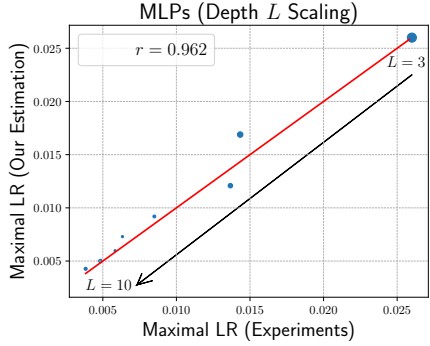

Figure 5: MLP networks of different depths on CIFAR-10. The x-axis shows the "ground truth" maximal learning rates found by our grid search experiments. The y-axis shows the estimated learning rates by our principle in equation 8. The red line indicates the identity. Based on the true maximal learning rate of the feedforward networks of $L = 3$, we scale up to $L = 10$. The radius of a dot indicates the variance over three random runs.

# E   MORE EXPERIMENTS

## E.1   IMAGENET

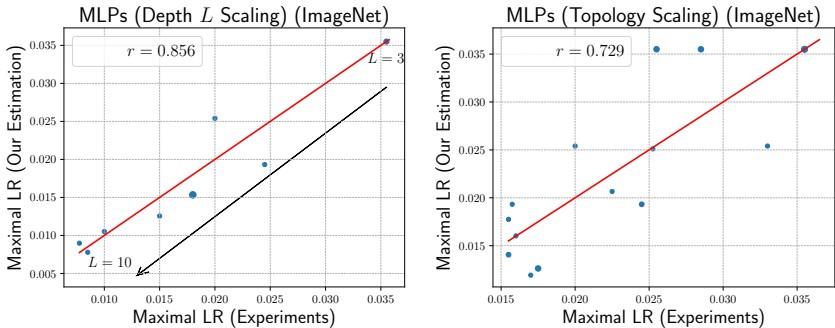

Figure 6: MLP networks (ReLU activation) of different depths (left) and graph topologies (right) on ImageNet Deng et al. (2009). The x-axis shows the "ground truth" maximal learning rates found by our grid search experiments. The y-axis shows the estimated learning rates by our principle in equation 8. The red line indicates the identity. The radius of a dot indicates the variance over three random runs.

Similar to Figure 5 and Figure 2, we further verify the learning rate scaling rule on ImageNet Deng et al. (2009). We scale up a vanilla feedforward network by increasing its depth (adding more Linear-ReLU layers) or changing its graph topology. As shown in Figure 7, our estimations achieve strong correlations with the "ground truth" maximal learning rates for both depth-wise scaling ($r = 0.856$) and topology-wise scaling ($r = 0.729$). This result demonstrates that our learning rate scaling principle is highly accurate across feedforward neural networks of different depths and across different datasets.

## E.2   THE GELU ACTIVATION

To demonstrate that our learning rate scaling principle can generalize to different activation functions, we further empirically verify MLP networks with GELU layers (on CIFAR-10). We scale up a vanilla feedforward network by increasing its depth (adding more Linear-ReLU layers) or changing its graph topology. Again, our estimations achieve strong correlations with the "ground truth" maximal learning rates for both depth-wise scaling ($r = 0.920$) and topology-wise scaling ($r = 0.680$). Moreover, for CNNs, our estimation can also achieve $r = 0.949$.

## E.3   THE $\mu$P HEURISTICS

We further compare the $\mu$P heuristics (Yang et al., 2022) with our architecture-aware initialization and learning rate scaling by training architectures defined in NAS-Bench-201 (Dong &

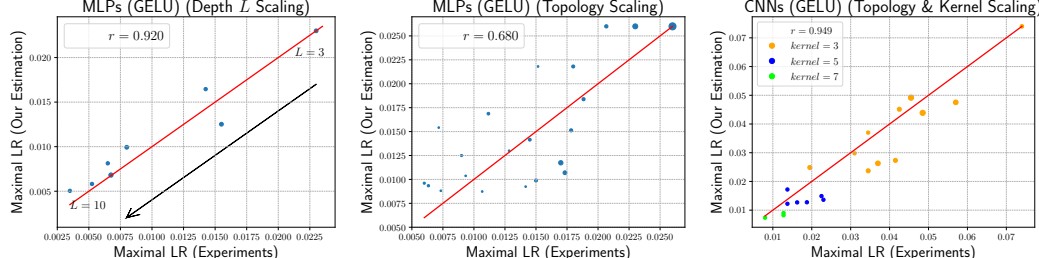

Figure 7: MLP networks with GELU activations of different depths (left) and graph topologies (middle), and CNNs (right), on CIFAR-10. The x-axis shows the "ground truth" maximal learning rates found by our grid search experiments. The y-axis shows the estimated learning rates by our principle in equation 8. The red line indicates the identity. The radius of a dot indicates the variance over three random runs.

Yang, 2020). We follow the usage from `https://github.com/microsoft/mup?tab=readme-ov-file#basic-usage` to set up the base model, and train with MuSGD. The learning rate is set as 0.1 following the original setting in NAS-Bench-201. From Figure 8, we can see that $\mu$P initialization and scaling strategy yields inferior results.

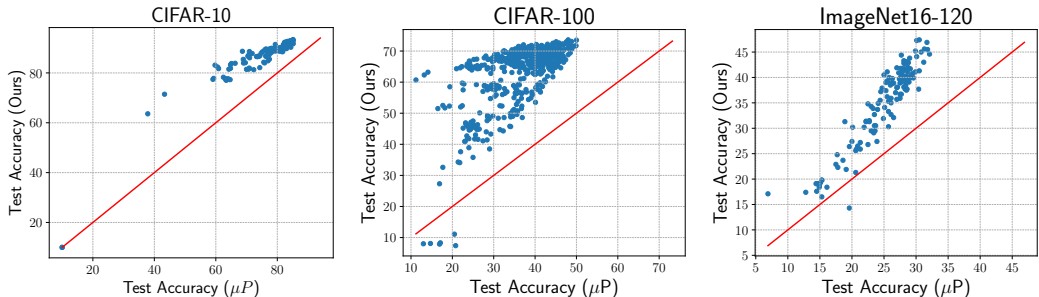

Figure 8: Comparison between the $\mu$P heuristics Yang et al. (2022) (x-axis) and our architecture-aware initialization and learning rate scaling (y-axis) on NAS-Bench-201 Dong & Yang (2020). Left: CIFAR-10. Middle: CIFAR-100. Right: ImageNet-16-120.

## E.4 NETWORK RANKINGS INFLUENCED BY RANDOM SEEDS

To delve deeper into Figure 4, we analyze how the random seed affects network rankings. Specifically, NAS-Bench-201 reports network performance trained with three different random seeds (seeds = [777, 888, 999]). For CIFAR-100, we created a plot similar to Figure 4 middle column, but it shows pairwise ranking correlations among those three random seeds. As shown in Figure 9, although we observe different ranking correlations between seeds, they are consistently higher (i.e., their rankings are more consistent) than those produced by our method. This confirms that changes in network rankings by our architecture-aware hyperparameters are meaningful, which can train networks to better performance.

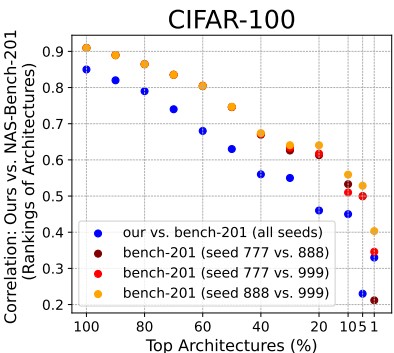

Figure 9: Comparison of network rankings influenced by our method versus random seeds. NAS-Bench-201 Dong & Yang (2020) provides training results with three different random seeds (777, 888, 999). Blue dots represent the Kendall-Tau correlations between networks trained by our method and the accuracy from NAS-Bench-201 (averaged over three seeds). We also plot correlations between random seeds in a pairwise manner. We compare networks' performance rankings at different top $K\%$ percentiles ($K = 100, 90, \cdots, 10, 5, 1$; bottom right dots represent networks on the top-right in the left column). This indicates that, although network rankings can be influenced by randomness during training, our method leads to significant changes in their rankings while still enabling these networks to achieve better accuracy.

