# OpenReview forum: "Principled Architecture-aware Scaling of Hyperparameters"
_ICLR.cc/2024/Conference — ICLR 2024 poster_

### Official Review · Reviewer_X2JZ · 2023-10-30

**Soundness:** 3 good
**Presentation:** 3 good
**Contribution:** 3 good
**Rating:** 6
**Confidence:** 4

**Summary:**

This paper provides valuable insights into the dynamic relationship among network architecture, initialization methods, and learning rates. It combines theoretical advancements with compelling empirical evidence, offering a precise characterization of how network architecture influences the interdependence of initialization strategies and the determination of optimal learning rates. The empirical evaluation is carried out on the NAS-Bench dataset, and the findings hold relevance not only for MLPs but also for CNNs with complex architectural designs. As a result, the implications of this research extend beyond the confines of deep learning theory, sparking meaningful reflections within the practical neural network design and NAS.

**Strengths:**

This paper delivers a compelling contribution that encompasses both theoretical and empirical dimensions. The clarity of its exposition, coupled with effective visual aids, greatly enhances comprehension of the concepts presented.

In terms of theoretical advancements, the authors expand upon the maximal-update (μP) scaling strategy initially introduced by Yang et al. in 2022. The authors delve into an investigation of how hyperparameters depend on various aspects of network architecture, including depth, connectivity patterns such as residual skips and long-range connections, kernel sizes, layer types, and convolution extensions.

The authors establish a link between learning rate scaling and layer depth for the first time, demonstrating its effectiveness over approaches that do not account for depth. Section E2 shows that the method also extends to GeLU neurons.

The authors showcase that the proposed scaled learning rates and initializations bring improvements in achievable accuracies across a broad spectrum of neural network architectures within the NAS-Bench search space. This outcome bears significant implications for the practical design of neural networks and the field of NAS, particularly considering the widespread adoption of NAS-Bench as a guiding benchmark.

The introduction of scaled learning rates and initializations yields a noteworthy consequence by narrowing the performance gap between "good" and "bad" architectures within NAS-Bench (Figure 4). They prompts questions about the ranking of architectures within NAS-Bench, prompting a critical reassessment of progress within the NAS field.

**Weaknesses:**

While this paper, on the whole, demonstrates a commendable effort, I encountered two noteworthy concerns while reading the experimental section:

-	The rankings of architectures may be influenced by various factors during training, including random seeds. It raises the question of how the authors determined that their revised learning rates and initializations would have the most significant impact on their observation of updated rankings.
-	It's recognized that NAS-Bench typically selects learning rates closer to those utilized by "good architectures" that are in practical use, reflecting an intentional bias towards favoring such architectures over "bad architectures." However, in Section 4.3, I could not discern evidence that the new learning rate/initialization approach improves upon the performance of the top-performing architectures, while it appears to only boost many of the "bad architectures." Given this, it raises the question of why this new learning rate/initialization strategy is relevant for practical NAS if it cannot enhance the performance of the best-found architectures.

**Questions:**

N/A

---

> ### Author Response · Authors · 2023-11-16
> **Thanks for your review!**
>
> We truly thank reviewer X2JZ's time and effort in reviewing our paper!
>
> > **Q1**: The rankings of architectures may be influenced by various factors during training, including random seeds. It raises the question of how the authors determined that their revised learning rates and initializations would have the most significant impact on their observation of updated rankings.
>
> Thank you for the question! We agree that numerous factors can influence architecture rankings.
>
> To delve deeper, we analyze how the random seed affects these rankings. Please kindly refer to Appendix E.4 in our updated paper.
>
> Specifically, NAS-Bench-201 reports network performance trained with three different random seeds (seeds = [777, 888, 999]). On CIFAR-100, we created a plot similar to the middle column of Figure 4, but it also displays pairwise ranking correlations among those three random seeds. This plot is included in Appendix E.4. In summary, although we observe different ranking correlations between seeds, they are consistently higher (i.e., correlations are more consistent) than those produced by our method.
> This confirms that changes in network rankings by our architecture-aware hyperparameters are *meaningful*, which can train networks to better performance.
>
> > **Q2**: I could not discern evidence that the new learning rate/initialization approach improves upon the performance of the top-performing architectures, while it appears to only boost many of the "bad architectures.
>
> Thank you! We would like to point out that in the left column of Figure 4, on both CIFAR-100 and ImageNet16-120, the improvement in the high-accuracy regime is either better than or as good as that in the low-accuracy regime.

---

### Official Review · Reviewer_nJvL · 2023-10-30

**Soundness:** 3 good
**Presentation:** 3 good
**Contribution:** 3 good
**Rating:** 6
**Confidence:** 4

**Summary:**

This paper proposes a method for optimizing hyperparameters in deep neural networks that takes into account the impact of neural architectures on hyperparameters. The authors characterize the dependence of initializations and maximal learning rates on the network architecture, including depth, width, conv kernel size, and connectivity patterns. They generalize their initialization and learning rates across MLPs and CNNs with sophisticated graph topologies. The authors verify their principles with comprehensive experiments and demonstrate that network rankings can be easily changed by better training networks in benchmarks with their architecture-aware learning rates and initialization.

**Strengths:**

-	The paper begins with a compelling motivation, shedding light on prevalent trends in the current literature landscape. It questions the evaluation of architectural choices and the often overlooked dependency on hyperparameters.
-	The paper provides thorough and well-supported proofs for its derivations related to initialization and maximal learning rates. It introduces a straightforward adjustment to weight initialization by scaling with the layers' in-degree, leading to notable performance enhancements downstream.
-	While the derivation draws substantial inspiration from previous works, it seems technically correct (and original to certain extent), and diligently cites these sources. Although maybe not the most groundbreaking, gaining a deeper understanding of how to determine initial learning rates for specific architectures holds significant importance.
-	The paper critiques the evaluation methodology employed in NAS benchmarks, uncovering significant shortcomings in the process in their experiments. The evaluation of NAS encompasses a wide range of comprehensive datasets, adding to the paper's robustness and relevance. There is a strong likelihood that the rules derived from this work could prove relevant and valuable for practitioners.
-	The paper exhibits excellent writing and organization, ensuring a smooth reading experience.

**Weaknesses:**

A few questions seem to hinder my understanding of this paper’s contributions, particularly with regard to the root causes behind the perceived limitations of the μP method.

First, the paper argue that their proposed method outperforms the μP initialization and scaling method introduced by Yang et al. in 2022, often achieving significantly better results. Noting that μP results were not included in the main paper and were instead instead deferred to a much later section E.3.

From a theoretical standpoint, Yang et al. (2022) primarily derived their scaling strategy for multi-layer perceptron (MLP) architectures in their main text, with additional derivations provided in Appendix L. On the empirical side, Yang et al. (2022) also presented experimental results on hyperparameter transfer for ResNet and Transformer architectures.

The present paper introduces a novel Directed Acyclic Graph (DAG) tool to derive "architecture-aware" hyperparameters. However, a fundamental question remains: What are the specific failure modes of the μP when it encounters intricate network topologies? Is the limitation attributed to a violation of Desiderata L.1, or are there other contributing factors at play?

Additionally, further clarification is needed regarding the assumptions made and the empirical comparisons drawn, especially across various architectural configurations. For example, it was not clearly mentioned until late, that the authors’ theory cannot apply to normalization layers.

**Questions:**

Same as Weakness

---

> ### Author Response · Authors · 2023-11-16
> **Thanks for your review!**
>
> We truly thank reviewer nJvL's time and effort in reviewing our paper!
>
> > **Q1**: What are the specific failure modes of the μP when it encounters intricate network topologies? Is the limitation attributed to a violation of Desiderata L.1, or are there other contributing factors at play?
>
> Thanks for your question! In the second paragraph of Section 2.2, we specifically discussed two core differences between $\mu$P and our method (and they are also reasons for $\mu$P’s pitfall in our scenario): 1) the non-trivial dependence of the learning rate on network depth, and 2) the complicated graph topologies of architectures. We will merge Figure 8 in Appendix E.3 into Figure 4.
>
> > **Q2**: Additionally, further clarification is needed regarding the assumptions made and the empirical comparisons drawn, especially across various architectural configurations. For example, it was not clearly mentioned until late, that the authors’ theory cannot apply to normalization layers.
>
> Thank you for pointing that out! In the conclusion section, we have clarified some of our limitations, including the existence of normalization layers. We will include additional discussions in our camera-ready version.

---

### Official Review · Reviewer_drLU · 2023-10-30

**Soundness:** 3 good
**Presentation:** 3 good
**Contribution:** 3 good
**Rating:** 6
**Confidence:** 4

**Summary:**

The paper aims to establish principles for initializing and selecting learning rates in neural network architectures characterized by directed acyclic graphs (DAGs), which can be highly irregular in structure. The authors propose an initialization method to maintain pre-activation variance during forward propagation and derive architecture-specific learning rates using a maximal update prescription. Experimental validation demonstrates their effectiveness and potential for improving neural architecture search benchmarks by enhancing network training and rankings.

**Strengths:**

-	The paper builds on previous research in hyperparameter and architecture search, aiming to establish a principled connection between weight initialization and learning rate choices with both MLP and CNN architectures.
-	It introduces an architecture-aware modified fan-in initialization scheme that preserves information flow through various graph topologies.
-	The paper analytically derives formulas for scaling learning rates based on architecture topology, specifically using the maximal update (μP) heuristic.
-	Experimental results demonstrate the superior performance of the proposed methods and highlight their potential to reshape network rankings in standard NAS benchmarks.
-	These findings suggest that implementing the proposed principles may lead to improved evaluations of NAS algorithms.

**Weaknesses:**

-	The clarity of the main body could be improved a lot, via providing a concise summary of main derivations, potentially by reducing the repetitive criticism of architecture search.
-	The experimental section raises several concerns. The strategy of finding a base maximal learning rate for one epoch may not be meaningful for practical training cycles, where learning rates follow complex schedules. The criticism of NAS for using "the same hyperparameters" overlooks the intricate learning rate strategies commonly employed.
-	The empirical results are also not entirely convincing. Figure 2 exhibits a weak correlation with questionable linearity and Figure 3 has a limited range of learning rates. The proposed improvements in Figure 4 seem to be primarily in lower accuracy regimes, raising doubts about absolute improvement on the top-performer architectures (which are of the most interest)
-	The discussion of prior art is limited, particularly regarding weight initialization and learning rates, and it's unclear how the proposed method advances over existing insights in the literature.

**Questions:**

See previous section of weakness.

---

> ### Author Response · Authors · 2023-11-16
> **Thanks for your review!**
>
> We truly thank reviewer drLU's time and effort in reviewing our paper!
>
> > **Q1**: The clarity of the main body could be improved a lot, via providing a concise summary of main derivations, potentially by reducing the repetitive criticism of architecture search.
>
> We have summarized our main derivations at the beginning of Section 3. We will further reduce the emphasis on the architecture search part and will update our draft accordingly in our camera ready.
>
> > **Q2**: The strategy of finding a base maximal learning rate for one epoch may not be meaningful for practical training cycles, where learning rates follow complex schedules.
>
> In Section 4.3, we strictly followed the cosine learning rate schedule used in NAS-Bench-201 to train different networks. Additionally, we discuss the practical implications of finding the base maximal learning rate for one epoch at the bottom of page 7.
>
> > **Q3**: Figure 2 exhibits a weak correlation with questionable linearity and Figure 3 has a limited range of learning rates. The proposed improvements in Figure 4 seem to be primarily in lower accuracy regimes.
>
> We respectfully disagree that a 0.838 correlation in Figure 2 is weak. Figure 3 spans an even wider range of learning rates than Figure 2. In the left column of Figure 4, for both CIFAR-100 and ImageNet16-120, the improvement in the high-accuracy regime is as good as, or better than, that in the low-accuracy regime.
>
> > **Q4**: The discussion of prior art is limited, particularly regarding weight initialization and learning rates, and it's unclear how the proposed method advances over existing insights in the literature.
>
> In the second paragraph of Section 2.2, we provide a detailed discussion of closely related works.
>
> Our work advances over existing insights: Previous studies have observed and developed heuristics about the relationship between learning rates and network structures. However, our core advantage over prior work lies in precisely characterizing the non-trivial dependence of learning rates on network depth, DAG topologies, and convolutional kernel size. This goes beyond previous heuristics and has practical implications.
>
> While our underlying principle may seem to echo patterns from earlier studies, such as using smaller learning rates for larger models, we provide the exact scale of this dependence (-3/2 for depth, and the summation of depths across the network’s graph structure). As confirmed by reviewers nJvL and X2JZ, our work offers a deeper understanding of how to determine initial learning rates for specific architectures and establishes a link between learning rate scaling and layer depth for the first time.

---

### Official Review · Reviewer_BP74 · 2023-10-31

**Soundness:** 3 good
**Presentation:** 3 good
**Contribution:** 3 good
**Rating:** 8
**Confidence:** 3

**Summary:**

This work provides an extension of $\mu$transfer to arbitrary network structures and an application of the resulting method to neural architecture search (NAS), specifically in order to make NAS benchmarks more comparable by ensuring all competing networks are given a better chance to train. The major contributions:

* An improved way to set initializations and layer specific learning rates to stabilise training over network scales
* Experimental results demonstrating that NAS benchmarks are flawed: making the training process more stable massively changes the results

Weights are initialized to be normally distributed with variance:
$$
\sigma^2 = \frac{C^{(l', l)}}{n} = \frac{2}{d_{in}^{(l')}}
$$
for an $n \times n$ weight matrix with $d_{in}^{(l')}$ in-degree (the number of connections to the layer from the previous layer).
Also, the final layer is scaled by $\propto \frac{1}{n^2}$.

Scale the layerwise learning rates according to
$$
\eta^{*} \simeq c \left( \sum_{p=1}^{P} L_p^3 \right)^{-1/2}
$$
where $c$ is a layer-independent constant and $L_{p}$ is the number of relu layers on path $p$.

**Strengths:**

Neural Architecture Search needs a way to verify that the networks being searched over are actually being trained sufficiently. The goal of this paper to standardise that process around a framework that should allow much more reliable results in this area of study.

I have not checked all the derivations but the theory appears to be correct. I am confident that I could verify the results if I had more time.

The contributions as presented are met:

* The initialization is provided, justified and works in the experiments
* The extension to $\mu$P is introduced clearly and also works in experiments
* Experiments demonstrate a key failing of NAS benchmarks and present a solution

The two limitations in the literature that this paper addresses are:

1. $\mu$P is not defined for arbitrary DAGs, only feedforward networks
2. Neural Architecture Search benchmarks are useless if we can't trust that the networks have been trained well

The presentation is good, explaining the key points of the $\mu$P paper in a brief way, better than the original paper.
The authors use an idiosyncratic system of emphasis, using both underlines and bold fonts on key points. This actually works quite well and I found the points being emphasised generally did earn more attention. The authors also use § in place of "Section", which I guess saved space, and works just as well.

**Weaknesses:**

The initialization, learning rate tuning and layer specific learning rates are introduced to maintain the update scale at $O(1)$ but the update scale during training doesn't appear to have been measured. It would be nice to see empirical verification that the method is working as intended. Although, I understand that the results in 4.1 and 4.2 both indicate that it is.

Given the equations above it is not easy for the reader to replicate the exact initialization and learning rate scaling required. Some pseudocode or a reference implementation would help a lot. It's been a problem for $\mu$P adoption as well, that practitioners find that there were ambiguities in the description that make it hard to implement in practice. For example, I don't know where the $c$ parameter is set and I'm not fully confident how to count the number of paths to produce $L_p$.

It is stated "the final layer weights are initialized with variance $1/n^2$ instead of $1/n$" on page 5, and I know this is from $\mu$P but it is never stated in this paper why.

Minor presentation issues:

1. "most designs of principles" in abstract doesn't make sense
2. Why is the MSE loss introduced in equation 6? The experiments are mostly classification results using a cross-entropy loss
3. Equation 8 contains $L_p$ but $L_p$ is only defined in the Appendix, it should be defined near the equation
4. Section 3.5 "speed-up" -> "speeds up"

**Questions:**

For each network architecture sampled in the NAS-Bench experiments, the network learning rate is first tuned at small scale. Was the learning rate also tuned on the networks being trained without this hyperparameter transfer method? This may already be in the experiments section or the Appendices and I may have missed it.

Why are the maximal LRs found by experiment in Section 4.1 and 4.2 not the same as those computed by the theory? Is there any way they could be brought closer together?

How was the experiment in Sections 4.1 and 4.2 performed? I guess you find the optimal learning rate at a small scale on a set of networks, then scale the network up using the method described in the paper to scale the initalization and learning rates, then perform a grid search to find the optimal learning rates to compare against?

---

> ### Author Response · Authors · 2023-11-16
> **Thanks for your review!**
>
> We truly thank reviewer BP74's time and effort in reviewing our paper!
>
> > **Q1**: Update scale during training.
>
> Thanks for the question! We further conduct experiments to verify the O(1) update scale for pre-activations.
>
> Specifically, using our settings in Section 4.1 (MLP with topology scaling), we measure the change in a network’s output $z^{(L+1)}$ (Equation 7) before and after the first gradient descent step (averaged over 10 minibatches, on CIFAR-10), and compare the norm of this change across different MLP architectures. Based on our analysis, the baseline (standard initialization plus a fixed learning rate) cannot preserve this O(1) update scale when scaling up networks and will also give a large variance across different architectures.
>
> Experiments show that $\mathbb{E}\left[\left(\Delta z_{i}^{(L+1)}\right)^2\right]$ for the baseline is 5.59 (std = 4.48), while our topology-aware initialization and learning rate yield a norm of 1.70 (std = 1.01). This result indicates that the success of our hyperparameter scaling is based on the preservation of the O(1) update scale across models.
>
> > **Q2**: Reference implementation ($C$ and $L_p$).
>
> * $C$ is initially set as 2 for ReLU in standard initialization, and is further updated to be architecture-aware (see Eq. 4) in our paper.
> * Calculating $L_p$ is straightforward: after representing the architecture as a computational graph, we derive its affinity matrix. Similar to Fig. 1, in this matrix, each element indicates the type of layer ($W$) used to connect two features ($x$ and $z^{(1)}, z^{(2)}, \cdots, z^{(L+1)}$); “0” indicates no connections (layers) between hidden features, “1” indicates a skip connection, and “2” indicates other parameterized layers (with ReLU activation). This affinity matrix reveals: 1) the number of paths (consecutive layers) from the input to the output (i.e., $P$); 2) the number of parameterized layers on each path (i.e., $L_p$).
> We will ensure comprehensive documentation in the README.md file in our released code.
>
> > **Q3**: The final layer weights are initialized with variance 1/n^2 instead of 1/n.
>
> We adhere to the variance of $1/n^2$ for the output layer, primarily to align with the desiderata outlined in Section J.2.1 of the referenced work ($\mu$P). This approach is chosen to preserve the O(1) size of pre-activations. It also accommodates the observation that "the input layer is updated much more slowly than the output layer" as mentioned in their Section 5.
>
> > **Q4**: Minor presentation issues.
>
> Thanks for these suggestions!
> * We derive our principle using the MSE loss, mainly for its clarity in analyzing the changes of the model’s output $z^{(L+1)}$ (Eq. 21).
> * $L_p$ is defined in Sec. 3.3 (after Eq. 7) and is also depicted in Fig. 1.
>
> > **Q5**: Was the learning rate also tuned on the networks being trained without this hyperparameter transfer method?
>
> The learning rate used in the baseline (x-axis in Fig. 4 left column) was not tuned. It was originally pre-determined in NAS-Bench-201.
>
> > **Q6**: Why are the maximal LRs found by experiment in Section 4.1 and 4.2 not the same as those computed by the theory?
>
> Maximal learning rates (LRs) found through experiments are not exactly the same as those predicted by our theory, mainly due to the more complicated training dynamics that are not fully captured by our analysis.
> Bring them closer together requires detailed analysis of the gradient descent beyond the first step.
> Despite this, our work demonstrates that analyzing only the first gradient descent step can still yield highly accurate learning rate predictions, with correlations exceeding 0.8.
>
> > **Q7**: How was the experiment in Sections 4.1 and 4.2 performed?
>
> Yes, our experiment pipeline is exactly the same as those three steps we stated at the beginning of Section 4.

---

> > ### Comment · Reviewer_BP74 · 2023-11-21
> >
> > Q1: The results you summarise here are promising, if these were presented in a table I would appreciate a comparison to $\mu$P heuristics. Otherwise, yes it appears that the update scale is being preserved. It may also be worthwhile using a very large batch size for this measurement to minimize SGD noise.
> >
> > Q2: I should probably have reviewed the code attached when I was reviewing this paper when making statements about example code that would be useful, I apologise.
> >
> > Q3: This was an issue with presentation but it is minor.
> >
> > Q4: I see where $L_p$ is defined now before the Appendix, on the line starting **Path Depth**.
> >
> > My concerns in the remaining questions are addressed, I will update my review.

---

### Author Response · Authors · 2023-11-16
**We thank comments, questions, and suggestions by all reviewers!**

We deeply appreciate the feedback and suggestions from all four reviewers. We’re pleased that **all four reviewers recognized** our strong motivation, the simple yet principled nature of our learning rate scaling, and our contributions to both theory and practice.
We address all questions and concerns in individual responses.

---

### Meta-Review · Area_Chair_ysJ5 · 2023-12-06

**Metareview:**

The paper proposes an extension of the original mup-paper (Tensor Programs V: Tuning Large Neural Networks via Zero-Shot Hyperparameter Transfer). The mup-paper provides heuristics so that hyperparameters are constant when scaling architectures in width. The paper under review extends this to a more general class of networks, and networks can also scale in depth. This is potentially very interesting and useful.

While the paper got acceptance ratings from all reviewers, some aspects on the method and the experimental evaluation are still unclear.

* One issue is that it is difficult to understand how the paper goes beyond mu-parameterization. Yes, the derivations are for more general directed acyclic graphs, but all simulations pertain to MLPs and CNNs to which mu-parameterization already applies to.

This and other related points lead to quite some confusion for me and the reviewers when reading the paper.

* Moreover, while the paper contains a comparison of the mup-heuristics to the paper's method in the appendix (E.3), it is unclear how the comparisons are carried out (e.g., for SGD or for Adam, etc), details are lacking.

**Justification For Why Not Higher Score:**

The paper's simulations and exposition lack in rigor, some statements made and potential improvements over existing methods (mu-parameterization) are difficult to understand or insufficient.

**Justification For Why Not Lower Score:**

The paper provides a potentially useful method for adapting hyperparameters when scaling models.

---

### Decision · Program_Chairs · 2024-01-16

Accept (poster)